# Acute Aquatic Toxicity to Zebrafish and Bioaccumulation in Marine Mussels of Antimony Tin Oxide Nanoparticles

**DOI:** 10.3390/nano13142112

**Published:** 2023-07-20

**Authors:** Ivone Pinheiro, Monica Quarato, Antonio Moreda-Piñeiro, Ana Vieira, Virginie Serin, David Neumeyer, Nicolas Ratel-Ramond, Sébastien Joulié, Alain Claverie, Miguel Spuch-Calvar, Miguel A. Correa-Duarte, Alexandre Campos, José Carlos Martins, Pilar Bermejo-Barrera, Marisa P. Sarriá, Laura Rodriguez-Lorenzo, Begoña Espiña

**Affiliations:** 1Water Quality Group, INL-International Iberian Nanotechnology Laboratory, Av. Mestre José Veiga, 4715-330 Braga, Portugal; ivone.pinheiro@inl.int (I.P.); monica.quarato@inl.int (M.Q.); ana.vieira@inl.int (A.V.); vmarisapassos@gmail.com (M.P.S.); laura.rodriguez-lorenzo@inl.int (L.R.-L.); 2Department of Analytical Chemistry, Nutrition and Bromatology, Faculty of Chemistry, University of Santiago de Compostela, 15782 Santiago de Compostela, Spain; antonio.moreda@usc.es (A.M.-P.); pilar.bermejo@usc.es (P.B.-B.); 3Centre d’Élaboration de Matériaux et d’Etudes Structurales (CEMES/CNRS), 29, rue Jeanne Marvig, 31055 Toulouse, France; virginie.serin@univ-tlse3.fr (V.S.); david.neumeyer@cemes.fr (D.N.); nicolas.ratel-ramond@cemes.fr (N.R.-R.); sebastien.joulie@cemes.fr (S.J.); claverie@cemes.fr (A.C.); 4TeamNanoTech/Magnetic Materials Group, CINBIO, Universidade de Vigo, Campus Universitario Lagoas Marcosende, 36310 Vigo, Spain; miguel.spuch.calvar@uvigo.es (M.S.-C.); macorrea@uvigo.gal (M.A.C.-D.); 5CIIMAR—Interdisciplinary Centre of Marine and Environmental Research, 4450-208 Matosinhos, Portugal; acampos@ciimar.up.pt (A.C.); jmartins@ciimar.up.pt (J.C.M.)

**Keywords:** Antimony Tin Oxide Nanoparticles, bioaccumulation, zebrafish embryo, marine mussels, aquatic toxicity

## Abstract

Antimony tin oxide (Sb_2_O_5_/SnO_2_) is effective in the absorption of infrared radiation for applications, such as skylights. As a nanoparticle (NP), it can be incorporated into films or sheets providing infrared radiation attenuation while allowing for a transparent final product. The acute toxicity exerted by commercial Sb_2_O_5_/SnO_2_ (ATO) NPs was studied in adults and embryos of zebrafish (*Danio rerio*). Our results suggest that these NPs do not induce an acute toxicity in zebrafish, either adults or embryos. However, some sub-lethal parameters were altered: heart rate and spontaneous movements. Finally, the possible bioaccumulation of these NPs in the aquacultured marine mussel *Mytilus* sp. was studied. A quantitative analysis was performed using single particle inductively coupled plasma mass spectrometry (sp-ICP-MS). The results indicated that, despite being scarce (2.31 × 10^6^ ± 9.05 × 10^5^ NPs/g), there is some accumulation of the ATO NPs in the mussel. In conclusion, commercial ATO NPs seem to be quite innocuous to aquatic organisms; however, the fact that some of the developmental parameters in zebrafish embryos are altered should be considered for further investigation. More in-depth analysis of these NPs transformations in the digestive tract of humans is needed to assess whether their accumulation in mussels presents an actual risk to humans.

## 1. Introduction

Nanotechnology is a fast-developing area that has introduced the massive use of engineered nanomaterials (ENMs) in our society. Through their characteristics, the ENMs improve the physicochemical properties and add extra functionalities to the material where they are incorporated. With a wide range of applications, ENMs can be used in different fields such as veterinary medicine [1,2], human health [3,4,5], aquaculture [6,7,8] and livestock production [9], environment [10,11], water treatment [12,13], and food [14,15,16]. The widespread use of nanomaterials leads to their release and inevitable presence in different aquatic systems [17]. The analysis of their environmental concentrations is very challenging, which makes it difficult to predict their behavior in aquatic environments [17,18,19,20].

Antimony tin oxide (ATO) nanoparticles (NPs) have as their main characteristics their transparency and good electrical conductivity. These particles have been used in the fabrication of bio/electrochemical devices [21,22,23] and solar cells [24], and conjugated with other materials can improve the treatment of wastewater [25,26] and air [27]. More recently, ATO NPs are being incorporated in smart windows [28] and studied as surface-enhanced Raman scattering (SERS) substrates [29]. Despite their industrial use, there are very scarce studies on their toxicity [30], and nothing about their environmental impact.

Water resources are considered one of the main sinks for nanomaterials in the environment [31]. In this study, we wanted to make a preliminary assessment of the potential impact of ATO NPs when reaching water environment (both freshwaters and marine waters) using two highly sensitive organisms that can provide insights on the acute toxicity and bioaccumulation potential of NPs: zebrafish and marine mussels, respectively.

Small size, low cost-maintenance, reduced housing requirements, quick development, and high genetic and physiologic similarity with humans [32,33] are some of the attractive features that make zebrafish (*Danio rerio*) a consensually accepted and used animal model to evaluate the potential risk of different xenobiotics present in water [34,35,36]. Following the 3Rs (replacement, reduction, and refinement) principle and ethically accepted by the European legislation [37], the Fish Embryo Acute Toxicity (FET) test [38] has been implemented as a reliable assay in the environmental toxicology field. The zebrafish embryos, for being transparent and presenting a fast and well-detailed described development [39], are excellent candidates for this type of test. The zebrafish embryotoxicity test is considered a robust and highly reproducible test with a good correlation with the results obtained for adults [40,41].

On the other hand, marine mussels *Mytilus* spp. are filter-feeding mollusks used for human consumption as seafood and an excellent contamination sentinel [42] due to their extraordinary filtration rate (2–3 L/h) [43]. In the last decades, marine mussels have been used as model organisms to evaluate their potential for bioaccumulation, bioconcentration, and biomagnification of suspended particulate material in seawater, such as ENMs or microplastics [44,45]. 

ATO NPs have been used for different applications; in particular, they have been incorporated in polymers to improve their optical properties and resistance to scratches. Their heavy metal content as well as their nano size makes them potentially hazardous materials for humans and the environment. Despite being produced at significant levels and incorporated in commercial products, the ecotoxicity of those nanoparticles has been disregarded. ATO NPs’ ecotoxicity and bioaccumulation in aquatic organisms are described in this manuscript for the first time. 

The ecotoxicity of ATO NPs was evaluated in this study by investigating its acute aquatic toxicity to adult zebrafish and their embryos. An Organization for Economic Co-operation and Development (OECD) test guideline 236 modified assay was used to register sub-lethal endpoints in the zebrafish embryo development able to provide insights into the action mechanism of toxicity and primary tissue targets. Furthermore, bioaccumulation experiments were carried out using *Mytilus* spp., studying whether ATO NPs get retained and concentrated in mussels and could consequently enter in the human food chain.

## 2. Materials and Methods

### 2.1. Nanoparticles Characterization

ATO nanopowder, NanoArc^®^ (Alfa Aesar, Kandel and Germany), 100% crystalline, non-porous, non-agglomerated particles, was purchased from Alfa Aesar (Kandel, Germany). The size and size distribution of the nanoparticles have been investigated using transmission electron microscopy (TEM), and the images were processed using a Tecnai F20 FEI operating at 200 kV, equipped with a spherical aberration corrector. The structures were determined with X-ray diffraction (XRD) measurements, performed on a Bruker D8 Advance diffractometer, equipped with a Cu anticathode, programmable divergence slits, soller slits on primary and secondary arms, and a Lynxeye position sensitive detector (Bragg Brentano geometry), and the identification of the phases used Rietveld refinement. The surface net charge of the pristine particles, specific surface area, and thermal behavior were studied with a Zetasizer Ver. 6.12 from Malvern Instruments Ltd. (Great Malvern, UK).

### 2.2. Test Suspensions Preparation

Artificial seawater was prepared by dissolving 3.5 g/L of commercial marine salt (Marine salt- ICA Basic Plus) into deionized water. ATO NPs stocks with a concentration of 10 g/L were prepared by dispersing the NPs in specific media according with the assay, i.e., freshwater or artificial seawater, using a sonicator probe with 50% amplitude and 30 s pulse on/5 s pulse off for 30 min. The ATO NPs were characterized in each media including ultrapure water as reference with Dynamic Light Scattering (DLS) and ζ-potential. The stock dispersions were diluted to reach a concentration of 100 mg/L. Both DLS measurements and ζ-potential were acquired with a SZ-100 device from Horiba at scattering angle at 173°.

For mussel exposure, the ENMs stock suspension was prepared using the same artificial seawater used for their maintenance. For each exposure test concentration, a stock suspension was prepared; stock I had a final nominal concentration of 1 mg/L in the aquarium, and stock II was 0.1 mg/L. Stock I was sonicated for 30 min and used to prepare the stock II which was sonicated for another 10 min. The stock solutions were freshly prepared every time before being applied in the aquaria.

For the adult zebrafish exposure, the suspensions were prepared so that the nominal test concentrations in the aquaria were 0.01, 0.1, 1, 10, and 100 mg/L NPs, and 0 mg/L as control. The suspensions were made in freshwater. To 100 mg/L of ATO NPs, the sonication time was 30 min, and 10 min more for the serial dilutions. Before the renewal of the medium, the suspensions were sonicated for 5 min in an ultrasonic bath (Elma, Elmasonic P, GE, Singen).

### 2.3. Mussel Maintenance and In Vivo Exposure Conditions 

Adult mussels, *Mytilus galloprovincialis* (shell length 7.11 ± 0.14 cm and width 3.4 ± 0.08 cm) were obtained already depurated from a local company (Falcamar; Labruge, Portugal). The acclimation period to the laboratory conditions was never less than one week. During this period, the water was periodically renewed, and temperature, salinity, conductivity, pH, and oxygen saturation were checked using a multiparametric probe (model HI98194, Hanna Instruments^®^, Póvoa de Varzim, PT). The temperature was maintained at 18 ± 2 °C and the photoperiod was at 14 h/10 h light/dark. The animals were fed every other day with commercial food for filter feeders (NT Labs, Tonbridge, UK).

Forty-six mussels were randomly distributed into aquariums, filled with artificial seawater, and individually aerated. For 28 days the mussels were exposed, in triplicates, to ATO NPs at the nominal concentrations of 0 mg/L as control, 0.1 mg/L, and 1 mg/L. During the exposure time, the mussels were fed every other day with living microalgae cultures of *Chlorella vulgaris* (10^5^ cell/mL). Once per week, full aquaria water was renewed and nanoparticles exposure was repeated by pre-mixing the freshly prepared ATO NPs suspensions with the microalgae, and subsequently poured into each aquarium. Two additional water renovations of 25% were performed per week. The mortality was verified every day during the full experiment (see Figure 1).

### 2.4. Sample Preparation and Quantification for Elemental and Nanoparticle Analysis with Inductively Coupled Plasma Mass Spectrometry (ICP-MS) 

Five mussels from each aquarium were collected after 1, 7, 14, 21, and 28 days of exposure to the ATO NPs. The mussels were de-shelled, the excess of water was removed, and the total soft tissue was frozen at −80 °C until further processing.

The alkaline digestion method was the selected technique for the ATO NPs extraction and quantification from dry mussels. Before digestion, the mussels were freeze-dried and the dry weight was recorded. The digestion approach involves the use of an aqueous solution of 10% (*v/v*) tetramethylammonium hydroxide (TMAH) followed by a 2 h bath sonication (Elma, Elmasonic P), and a centrifugation step (2500× *g*, 20 min). A second TMAH digestion was then performed with the obtained pellet for 1 h. The mixture was centrifuged at 2500× *g* for 20 min and redispersed in 1% (*v/v*) sodium dodecyl sulphate (SDS) to improve the NP’s separation from proteins aggregates/organic matter. In the last step, 15 mL of hydrogen peroxide (30% *v/v* H_2_O_2_) was gradually added to the suspension while heating at 70 °C under stirring, reaching a final pale-yellow color, evidencing the complete organic fraction digestion. The method recoveries were calculated using artificially spiked samples with 0.5, 1, 1.5, or 2 mg/L of ATO NPs (Appendix A). The solution was then cooled down and filtered using a 5 µm pore size cellulose acetate (CA) filter. Standards were prepared by spiking the mussels with ATO NPs in the range of 0–2 mg/L. Blanks were prepared to correct the results from the potential procedure’ contamination.

The total antimony and tin concentration, as well as the ATO NPs quantification, was performed using a NexIon 2000 ICP-MS (Perkin Elmer, Waltham, MA, USA). ATO NPs assessment was carried out working in the single particle mode (sp-ICP-MS, Appendix A)).

### 2.5. Fish Acute Toxicity (FAT) Test 

#### Zebrafish Animal Maintenance and Acute Exposure

FAT tests were performed according to OECD test guideline 203. In short, adult zebrafish were acclimated to the laboratory conditions for at least one week. Water quality was controlled, and the following physical–chemical parameters were measured: ammonium, nitrites, pH, temperature, conductivity, and dissolved oxygen. The temperature was 27 ± 1 °C and the photoperiod 14 h/10 h light/dark. The animals were fed daily with commercial dry flakes (TetraMin, Tetra, Germany). 

Ten fish, both genders, were randomly distributed in a 10 L aquarium, individually aerated and heated, and exposed in triplicates for 96 h to the test nominal concentrations previously indicated. During this period, the animals were fasting, and mortality was checked daily (see Figure 2).

### 2.6. Fish Embryo Acute Toxicity (FET) Test

#### 2.6.1. Zebrafish Spawning and Eggs Assortment

Wild-type zebrafish breeders were housed in a 50 L tank of dechlorinated and aerated freshwater, connected to a unidirectional recirculation flow pump, coupled to mechanical and biological filters, and acclimated at 25 ± 1 °C, for 14 h/10 h light/dark. Twice a day, parental specimens were fed *ad libitum* with commercial flakes TetraMin (TetraMin, Tetra, Germany), and supplemented every two days with (live) *Artemia* spp. nauplii (Ocean Nutrition, Newark, CA, USA).

A 30 L mesh bottom cage covered with an artificial substrate (marbles) filled with dechlorinated and aerated freshwater at 28 ± 1 °C, under the same photoperiod as the housing tank was used to host the breeding mates, with a male-biased sex ratio group. 

Spawning events occur at light onset among sexual mates, conducting external fertilization. The eggs were collected and submitted to a series of washing steps, and the viable eggs were separated from unfertilized or dead zygotes under a stereo microscope, taking into account their distinctive optical transparency. Pre-heated freshwater used in experiments was pre-filtered on a Millipore Stericup-GP sterile vacuum system, coupled with a 0.22 µm pore size polyethersulfone membrane. All zebrafish eggs treated per experiment derived from the same laying episode. Only egg clusters showing a fertilization rate greater than 90% were used.

#### 2.6.2. Zebrafish Embryotoxicity Test

ATO NPs’ embryotoxic effects were assessed at different developmental stages of zebrafish (as referred at Kimmel et al., 1995), in an adaptation of OECD test guideline 236. Ten viable zygotes per replicate (well) were arbitrarily transferred to a 24-well plate, and waterborne exposed (in a semi-static regime) to the serial diluted nominal concentrations 0, 0.01, 0.1, 1, 10, and 100 mg/L of ATO NPs, for 80 h post-fertilization (hpf). Pre-filtered freshwater was defined as the experimental control. Four replicates of ATO NPs test concentrations were screened at two independent experiments. Two mL per well (replicate) were used (see Figure 2).

ATO NPs test concentrations were first dispersed as detailed before, and the pH was verified at the lowest and highest nominal concentrations to assure a tolerable environment to zebrafish normal embryonic development [39]. ATO NPs test concentrations were pre-heated at 26 ± 1 °C each day before the assay, and FET plates were exposed to the same photoperiod as the spawners’ tank. One experiment was acknowledged as “valid” to an embryonic lethality threshold set at 25% to experimental control.

At 8, 32, 56, and 80 hpf, different age-correspondent embryonic developmental events were investigated, e.g., at 8 hpf, epibolic arc perimeter, heart rate for 32 and 56 hpf, and at 80 hpf, occurrence of burst swimming. Idiosyncratic morpho-physiological and behavioral modifications occur at these early life stages [39], thus permitting the detection of premature signals of toxicity. To assess the ATO NPs-associated cardiotoxic effects, twenty zebrafish embryos per test concentration were randomly selected, and the heart beating was counted for 10 s. A detailed morphometric analysis of the chorion, yolk, eye, and pupil (respective per hpf) was performed, using ImageJ (v.1.53e), and their specific developmental characteristics were photographed using a Nikon Eclipse Ts2 inverted microscope coupled to a Nikon DS-Fi3 camera. Several sub-lethal endpoints were assessed: atypical hatching events, anomalous cellular masses, structural aberrations, irregular movements, delayed traits, and defective growth. Survival was checked at all hpf. The abovementioned test variables were particularly selected given their presumed involvement at soundly conserved embryonic developmental processes among vertebrates.

#### 2.6.3. FET Test Data Analysis

##### Statistical Assessment

All assumptions were met before the data statistics, namely normality, using the Shapiro–Wilk test and homogeneity of variances, using Levene’s test. 

To investigate the ATO NPs exposure effects on zebrafish embryos’ spontaneous movements, hatching rate, and cumulative survival, a chi-square (χ^2^) test was conducted considering the observed values for each test (nominal) concentration. The null hypothesis of “no differences among groups” was considered to outline the expected values (respectively, the average percentage of involuntary motion events at 32 hpf, chorion extruding rate at 56 hpf, or total embryonic survival at 80 hpf, of all treatments). 

One-factor ANOVA (six levels: 0, 0.01, 0.1, 1, 10, and 100 mg/L of ATO NPs) was used to test the effect on zebrafish embryos’ epiboly, head-trunk angle, and burst swimming.

ANCOVA analysis was performed to avoid biases related to covariates to assess the ATO NPs effect on zebrafish yolk volume (egg volume was used as co-variable) and pupil surface (eye surface was used as co-variable).

In order to investigate the impact of ATO NPs exposure on zebrafish embryonic heart rate, a nested ANOVA [two factors: test nominal concentration (six levels: 0, 0.01, 0.1, 1, 10, and 100 mg/L of ATO NPs; and developmental ages (two levels: 32 and 56 hpf)] was applied.

Post hoc comparisons were run using Student–Newman–Keuls (SNK). A *p* value of 0.05 was set for significance testing. Analyses were performed in STATISTICA (TIBCO software, v. 14).

### 2.7. Ethics Statement

All experiments that implicated the use of adult specimens of wild-type zebrafish were conducted at CIIMAR, an Interdisciplinary Centre of Marine and Environmental Research (Matosinhos, Portugal) that presents a dedicated bioterium of aquatic organisms (BOGA) certified by the Directorate General of Food and Veterinary (DGAV), the Portuguese National Authority for Animal Health, issued under Article 21º, Decree-Law 113/2013 of 7th August. At BOGA, FAT tests were run in agreement with OECD guideline 203 and subjected to a prior ethical review by CIIMAR’s Ethical Committee and Animal Welfare Body (ORBEA), in compliance to Directive 2010/63/EU on the protection of animals used for scientific purposes, subscribing to the principles, rules, and procedures of the European Code of Conduct for Research Integrity. At CIIMAR, Laboratory Animal Sciences training and DGAV certification are mandatory for researchers to perform animal testing involving vertebrates.

Tests on fish embryonic stages were conducted at the International Iberian Nanotechnology Laboratory (INL) (Braga, Portugal), as an adaptation to FET test (OECD test guideline 236) as defined at the Council of Europe Directive 86/609/EEC on protection of experimental animals, setting the regulatory limit of exposure at the free-living stage (that is, at the end of embryogenesis). As the last endpoint tested preceded this developmental age, the provisions of the directive do not apply, and therefore, a ratified ethical consent was not required.

At FET and FAT experiments, 100 mg/L was considered as the highest test (nominal) concentration of ATO NPs, according to OECD test guideline 203 which states limit for testing of chemicals at this range.

## 3. Results

Commercial ATO NPs used in this study are highly polydisperse, as demonstrated by the data obtained using DLS and TEM. Although the mean hydrodynamic diameter is 107 nm, a broad distribution of sizes can be found, and a large proportion of small NPs around 5 nm diameter are present (Figure 1).

Additionally, the XRD pattern analysis after Rietvelt refinement shows that the composition of the NPs is actually Sn_0.6_Sb_0.4_O_2_ in crystallites of 16.8 nm. ATO NPs present a ζ-potential of −32 ± 3.5 mV in a large distribution, a specific surface area (BET) of 143 ± 5 m^2^ g^−1^, a mean pore diameter of 8.48 nm, and a total pore volume of 0.3039 cm^3^·g^−1^ (Figure 1).

It is well know that NPs can aggregate in media containing high salt concentration such as seawater and that the formation of NP aggregates can influence their behavior in the media, and therefore, their availability to interact with specific organisms (here zebrafish adults and embryos, and mussels) [46,47]. Therefore, the colloidal stability of ATO NPs in artificial seawater and freshwater was tested. The ATO NPs remained colloidally stable in freshwater, while tending to aggregate in seawater (Table 1). Clearly, the difference in the ionic strength of each medium (i.e., the concentration of salts: <1 ppt in salinity for freshwater versus 35 ppt salinity for seawater) had a strong impact in the colloidal stability; the higher ionic strength presented in seawater provoked the aggregation of the ATO NPs due to screening of the electrostatic interaction, suppressing the stabilizing effect of the electric double layer (EDL). The increase in the hydrodynamic diameter observed in seawater is in agreement with the almost null ζ-potential measured in this medium. On the other hand, the slightly variation both hydrodynamic size and ζ-potential of ATO NPs in fresh water in comparison with ultrapure water (Table 1) can be attributed to the variation in the composition of EDL since different ions present in freshwater can be adsorbed to the NP surface modified the diffuse layer [48].

No mortality was observed either in the control group or at any of the test concentrations with ATO NPs in the FAT test (Appendix A). Likewise, no significant mortality was found when exposing zebrafish embryos to the ATO NPs during the first 80 h of development (Figure 2A). However, some sub-lethal effects were observed; heart beats of embryos at 32 hpf were significantly less at the highest (nominal) concentration tested (Figure 2B, 100 mg/L), and the number of embryos presenting spontaneous movements at the same hpf were also less and decreased dose-dependently until 10 mg/L (Figure 2C). 

Other morphometric parameters evaluated, such as eye surface and volume, head-trunk angle, yolk extension, or total body length, remained unaltered (see Appendix A).

The concentrations of 0, 0.1, and 1 mg/L were the selected nominal concentrations to test the ATO NPs accumulation into the bivalve mollusks during 28 days of exposure, performing four different sampling every seven days. 

For each time point, six different mussels from three independent replicates were analyzed for total Sb and Sn quantification after being subjected to the TMAH alkaline digestion (see details in the Materials and Methods section). The low amount of Sb (11%) contained in the particles does not allow the NPs determination, and thus only Sn was quantified in its nano form using sp-ICP-MS.

Sn NPs concentration shows a time and dose-dependent accumulation trend, reaching the highest level of 2.31 × 10^6^ ± 9.05 × 10^5^ NPs/g of dry weight in the last time point, after 28 days of exposure (Figure 3A). The high SD value could be attributed to the considerable number of dilutions performed prior to sp-ICP-MS analysis. Taking a look at the dissolved Sn concentration measured in the Nano configuration, the relatively low amount indicates that the technique is not responsible for NPs dissolution, and this could also be confirmed by the averaged particles’ size calculation (Figure 3B,C). It is also interesting to point out that there is no trend in dose or time-dependency on the ionic concentration, indicating that there is no significant dissolution of the ATO NPs in the mussel’s tissue. The presence of some NPs observed in the control condition could suggest that NPs that contain Sn in their composition were already accumulated into mussels at the starting time of the experiment. 

As concerned with the total Sb and Sn recovery, the measured element amount increased related to the time of exposure, especially at higher concentrations (Figure 4), which is in agreement with the results obtained using the NPs with sp-ICP-MS.

## 4. Discussion

ATO NPs clearly aggregate in seawater. This destabilization is due to the decrease in ζ-potential from—32 mV to almost null surface charge (Table 1), most probably due to the presence of a high ionic strength in seawater, which decreases the repulsion between NPs, caused by the EDL [48]. This aggregation of ATO NPs seems to occur because only electrostatic repulsions were presented since no additional stabilizer (e.g., polymer) was used. However, this artificial seawater did not contain natural (NOM) or particulate organic matter, nor any of the other substances that are secreted or excreted by living organisms into the water. NOM can increase ENMs’ stability, extending their residence time in the water column and consequently increasing the exposure of aquatic biota, including benthic organisms. In addition, polymeric substances secreted by aquatic microorganisms and bivalve mollusks, such as extracellular polymeric substances (EPS), polysaccharides, and proteins, may induce aggregation/agglomeration, acting as chelating agents to bind and stabilize ENMs dispersion [49].

ATO NPs induce an effect on the zebrafish embryo neuromotor system that does not advance to fatal consequences in the following development, as the heart beats recover at 56 hpf. Spontaneous movements come from a primitive spinal network [50] and are triggered jointly with the proteolytic fragmentation of the chorionic membrane inner layer [51], to assist the egg-shell rupture towards a successful hatching event. On the other hand, heart beating is mediated through secondary neuromotor circuits, responsive to the inputs of a mature hindbrain, requiring a complex arrangement of functional transmitter stripes at later stages of development [52]. Other ENMs have been recently found to alter those parameters in zebrafish embryos [53]. Thus, a careful analysis of chronic or sub-acute exposures to ATO NPs should be considered.

In the only previous study found regarding their toxicity, ATO NPs resulted quite innocuously to differentiated MucilAirTM bronchial epithelial cells, a well-stablished model for human pulmonary toxicity. Only low cytotoxicity was observed after 24 h of air-liquid interface exposure to aerosolized suspensions of 10 mg/mL ATO NPs, which was recovered in longer times of exposure [30]. Other cytotoxicity studies on pulmonary A549 cells have previously shown long-term exposure toxicity of Sb_2_O_3_ NP displaying an EC50 of 22 mg/mL [54], and reactive oxygen species (ROS) generation, expression of heme oxygenase 1 (HO-1) gene, and DNA damage by indium-doped SnO_2_ (ITO) [55]. These doses highly exceed the ones environmentally relevant, so no correlation can be established with the data obtained in this study. 

Marine mussels are filter-feeding mollusks that retain particles usually in a range between 5 and 35 μm in diameter. The percentage of retention decreases with the size and the smallest particles that are effectively retained (100%) are close to 7 μm [56]. Maximum retention efficiency was reported at 30 to 35 μm. The average hydrodynamic diameter of ATO NPs in artificial seawater was close to 10 µm (Table 1), making feasible their efficient retention in the mussels’ digestive system. 

Despite the fact that there is an existing time-dependent accumulation of ATO NPs in the mussels and that available studies confirm the digestive gland being the main organ for ENMs accumulation in bivalve mollusks [49], we could not find NPs in the digestive gland and mantle tissue analyzed using TEM, as the number of ATO NPs is not enough to allow any detection. Usually, only when the number of particles accumulated is higher than 1 × 10^8^ NPs/g of tissue there is a fair possibility to find them. In any case, tissues from mussels exposed to the ATO NPs did not show any ultrastructural difference when compared to the control (Appendix A). However, recent studies such as the one by Sun et al. [57] found between 2.1 × 10^6^ and 8.4 × 10^6^ particles/mg of Ti-containing NPs in marine mussels, suggesting that marine shellfish may be a significant sink for Ti-containing NPs. 

## 5. Conclusions

Commercial ATO NPs seem to be quite innocuous to aquatic organisms; no acute toxicity was registered on zebrafish adults or eggs/embryos. However, the significant alteration of heartbeat and spontaneous movements observed deserve a deeper investigation. Additionally, a slight bioaccumulation was observed in mussels (2.31 × 10^6^ ± 9.05 × 10^5^ NPs/g), indicating that ATO NPs are not a potential concern to human health by seafood consumption. 

## Data Availability

The data presented in this study are available on request from the corresponding author. The data are not publicly available due to internal privacy policy.

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
