# Peer review of "Acute Aquatic Toxicity to Zebrafish and Bioaccumulation in Marine Mussels of Antimony Tin Oxide Nanoparticles"

_nanomaterials, 2023, doi:10.3390/nano13142112_

Round 1

Reviewer 1 Report

The manuscript Nanomaterials-2431995 presents a study about the ecotoxicity of Antimony Tin Oxide Nanoparticles (ATO NPs). Despite ATO NPs industrial use, there are very scarce studies on toxicity and nothing about their environmental impact. In this context, the main quest addressed by this research is the evaluation for the first date of the acute toxicity exerted by commercial Sb2O5/SnO2 (ATO) NPs in adults and embryos of zebrafish (Danio rerio).In order to register sub-lethal endpoints in the zebrafish embryo development, an Organization for Economic Cooperation and Development (OECD) test guideline 236 modified assay was used, that provide insights on action mechanism of toxicity of ATO NPs and primary tissue targets. The possible bioaccumulation of these NPs in the aqua cultured marine mussel Mytilus sp. was also studied.  As the authors have found only one previous study regarding ATO NPs toxicity on differentiated MucilAirTM bronchial epithelial cells, a well-stablished model for human pulmonary toxicity, I consider that the present study is original, interesting and it is relevant for the field. By using a very performant technique (sp-ICP-MS), a quantitative analysis was performed in the present study to assess the possible bioaccumulation of these NPs in the aqua cultured marine mussel Mytilus sp.  In the previous papers with the same subject, NPs were not be found in the digestive gland and mantle tissue analyzed by TEM, as the number of ATO NPs is not enough to allow any detection.

The manuscript is well written, the text is clear and easy to read. It is scientifically sound and the experimental design is appropriate to test the hypothesis and I consider that the authors do not have to improve methods or experiments. The manuscript’s results are reproducible based on the details given in the methods section. The figures/schemes and tables are appropriate, show properly the data and they are easy to interpret and understand. The data are interpreted appropriately and consistently throughout the manuscript. The conclusions are consistent with the evidence and arguments presented.

I believe the manuscript matched the scientific scope of NANOMATERIALS and can be considered for publication without corrections.

I believe the manuscript matched the scientific scope of NANOMATERIALS and can be considered for publication without corrections.

Author Response

We highly appreciate the positive feedback by the reviewer. Thank you for your consideration.

Reviewer 2 Report

The aim of this manuscript by Ivone Pinheiro et al. is to assess the acute aquatic toxicity to Zebrafish and bioaccumulation in marine Mussels of commercial ATO NPs. Nanoecotoxicology is a very important topic. However, the written structure and English of this manuscript are too weak, and it only involves simple toxicity evaluation without any toxicological mechanisms study. Taken together, I CANNOT recommend this paper for publication in the journal NANOMATERIALS.

Major concerns:

1.        The authors employed two model organisms to assess ATO NPs toxicity, why? Zebrafish is also a suitable animal model for bioaccumulation detection.

2.        Since the authors used two types of model animals (Zebrafish and Mussels), both should be mentioned in your Scheme 1.

3.        Please indicate your dose selection criteria. 0.1-1 mg/L of ATO NPs is a very high dose in the real environment. Is there any relevant literature indicating this?

4.        How to explain the differences of hydrodynamic diameter, PDI, and zeta potential? Please indicate this.

5.        I strongly recommend that the author should re-draw their figures. The differences between Figure 3 and other two figures make them appear to come from different articles.

6.        The discussion part seems a repeat of your results part. I can’t see any discussion.

Minor concerns:

1.        Line 231, ANCOVA? Line 178, OC?

2.        Sb2SnO5 throughout the text should be Sb2SnO5. Please pay attention to the writing standards.

3.        Line 126, Chlorella vulgaris proper noun should be italicized. Please carefully review your entire text.

4.        Figure 1. Please standardize the labeling ABCD.

Extensive editing of English language required.

Author Response

The aim of this manuscript by Ivone Pinheiro et al. is to assess the acute aquatic toxicity to Zebrafish and bioaccumulation in marine Mussels of commercial ATO NPs. Nanoecotoxicology is a very important topic. However, the written structure and English of this manuscript are too weak, and it only involves simple toxicity evaluation without any toxicological mechanisms study. Taken together, I CANNOT recommend this paper for publication in the journal NANOMATERIALS.

Major concerns:

  1. The authors employed two model organisms to assess ATO NPs toxicity, why? Zebrafish is also a suitable animal model for bioaccumulation detection.

Reply:  We agree with the reviewer in the fact that zebrafish is a suitable model and has recently been used to evaluate bioaccumulation of chemicals and nanomaterials. However, we decided to select marine mussels as model for bioaccumulation instead because Mytilus spp., are filter feeding molluscs used for human consumption as seafood and an excellent contamination sentinel, due to their extraordinary filtration rate (2-3L/h) [41]. In the last decades, marine mussels have been used as model organisms to evaluate their potential of bioaccumulation, bioconcentration and biomagnifi-cation of suspended particulate material in seawater, such as ENMs or microplastics [42,43]. Marine mussels are recognised models for nanoparticles’ bioaccumulation assays (https://doi.org/10.1016/j.envpol.2020.114043 and highly relevant in human oral exposure due to its consumption as seafood

  1. Since the authors used two types of model animals (Zebrafish and Mussels), both should be mentioned in your Scheme 1.

Reply:  Thank you for the suggestion. For a better clarity we added a new scheme (scheme 2) to show the exposure conditions for the zebrafish toxicity assays.

  1. Please indicate your dose selection criteria. 0.1-1 mg/L of ATO NPs is a very high dose in the real environment. Is there any relevant literature indicating this?

Reply:  To the best of our knowledge, there is no specific literature on environmental concentrations (either predicted or real) for this type of nanoparticles yet. Given the lack of information, the selection of the exposure concentrations was based on worse-case scenario exposure estimation according to the maximum environmental concentrations found for TiO2 nanoparticles. The predicted environmental concentration (PEC) for TiO2 nanoparticles in surface waters are in the order of µg/L. Our lower concentration tested was aligned with this concentration range. However, due to high accumulation of NPs prone to sediment in sludge, and the resuspension process together with the increased use of the NPs, it is estimated that concentrations could raise close to the wastewater treatment plants effluents up to mg/L levels (https://doi.org/10.1016/j.scitotenv.2022.153866).  

  1. How to explain the differences of hydrodynamic diameter, PDI, and zeta potential? Please indicate this.

Reply:  The differences in the physicochemical properties of the ATO NPs are due to the different composition of the media where these NPs were dispersed. We have shown in the manuscript the characterization in both media used for the assays: artificial freshwater for ZET assay and artificial seawater for bioaccumulation assay. We have also included the characterization in ultrapure water as reference.

We did not describe appropriately this in the method section. Therefore, we have modified this section including a better description of the characterization methods in the different media.

ATO NPs stocks with a concentration of 10 g/L were prepared by dispersing the NPs in specific media according with the assay, i.e. freshwater or artificial seawater, using a sonicator probe with 50% amplitude and 30 s pulse on / 5 s pulse off for 30 min. The ATO NPs were characterized in each media including ultrapure water as reference by Dynamic Light Scattering (DLS) and zeta potential. The stock dispersions were diluted to reach a concentration of 100 mg/L. Both DLS and Zeta Potential were acquired with a SZ-100 device from Horiba at scattering angle at 173°.

Moreover, we have included a better discussion:

It is well know that NPs can aggregate in media containing high salt concentration such as seawater, and that the formation of NP aggregates can influence their behaviour in the media, and as a consequence, their availability to interact with specific organisms (here zebrafish adults and embryos, and mussels) [44,45]. Therefore, the colloidal stability of ATO NPs in artificial seawater and freshwater was tested. ATO NPs remained colloidally stable in freshwater, while tended to form aggregates when in seawater (Table 1). Clearly, the difference in the ionic strength of each medium (i.e. the concentration of salts: < 1 ppt in salinity for freshwater versus 35 ppt salinity for seawater) had strong impact in the colloidal stability: the higher ionic strength presented in seawater provoked the aggregation of the ATO NPs due to screening of the electrostatic interaction, suppressing the stabilizing effect of the electric double layer (EDL). The increase of the hydrodynamic diameter observed in seawater is in agreement with the almost null zeta potential measured in this medium. On another hand, the slightly variation both hydrodynamic size and zeta potential of ATO NPs in fresh water in comparison with ultrapure water (Table 1) can be attributed to the variation in the composition of EDL since different ions present in freshwater can be adsorbed to the NP surface modified the diffuse layer (https://doi.org/10.1039/C4CS00487F).  

  1. I strongly recommend that the author should re-draw their figures. The differences between Figure 3 and other two figures make them appear to come from different articles.

Reply:  We thank the reviewer for the suggestion. We modified figure 2 to look more similar in style to figures 4 and 5.

  1. The discussion part seems a repeat of your results part. I can’t see any discussion.

Reply:  The scarcity of literature with ATO NPs reduces the possibility of direct comparison. However, we included some more discussion as suggested based on results with other NPs.

Minor concerns:

  1. Line 231, ANCOVA? Line 178, OC?

Reply:  ANCOVA is a general linear model that combines ANOVA and regression analysis. ANCOVA evaluates whether the means of a dependent variable (e.g. yolk volume; pupil volume - respectively) are equal across levels of a categorical independent variable (treatment), while statistically controlling for the effects of other continuous variables that are not of primary interest, known as covariates (e.g. egg volume; eye volume - respectively) or nuisance variables. Mathematically, ANCOVA decomposes the variance in the dependent variable into variance explained by the covariates, variance explained by the categorical IV, and residual variance. In basic terms, the ANCOVA examines the influence of an independent variable on a dependent variable while removing the effect of the covariate factor.

OC refers to °C. It has been corrected in the manuscript.

  1. Sb2SnO5 throughout the text should be Sb2SnO5. Please pay attention to the writing standards.

Reply:  We apologise for the typo. There was a change in format when transforming the manuscript to pdf. We will re-check it.

  1. Line 126, Chlorella vulgaris proper noun should be italicized. Please carefully review your entire text.

Reply:  We apologise for the typo. There was a change in format when transforming the manuscript to pdf. We will re-check it.

  1. Figure 1. Please standardize the labeling ABCD.

Reply:  We thank the reviewer for the suggestion. The labelling was standardized across the figures.

Reviewer 3 Report

Herein, authors investigated the acute aquatic toxicity and bioaccumulation of antimony tin oxide (ATO) nanoparticles by using zebrafish and marine mussels. The data demonstrated that the ATO had little influence on the zebrafish and barely accumulate in the marine mussel, indicating that the ATO NPs may not be a potential concern to human health by seafood consumption. I find the manuscript to be interesting; however, it appears that the level of novelty may not be as significant as expected. I would recommend the authors to provide a more comprehensive discussion on the practical implications of their work. By emphasizing the practical significance and potential applications, the manuscript could still offer value to the readership and warrant consideration for acceptance.

Some issues should be addressed.

1.     The resolution of Figure 1 should be improved to provide readers with clearer images for better interpretation.

2.     Higher magnification of ATO NPs should be provided in Fig. 1a. What is the scale bar of Fig. 1a and 1b?

3.     What is synthetic seawater? The components of artificial seawater should be introduced in the Materials and methods section.

4.     In Figure 3, what did time point represent in x axis? Regarding the data presented in Figure 3, it would be beneficial for readers to have a clearer understanding of the time points referred to on the x axis. It is important to provide clear labeling of x axis.

5.     In line 138, 146,270 and 272, the number should be written in subscript.

6.     For in vivo study of the acute toxicity, the ATO NPs seemed less lethal to the zebrafish adults or embryos. Besides, more experiments should be conducted to confirm that point, such as morphological analysis, genetic profiles as well as histological analysis.

The manuscript should be carefully checked and revised to avoid the expression, grammar and spelling errors. 

Author Response

Some issues should be addressed.

  1. The resolution of Figure 1 should be improved to provide readers with clearer images for better interpretation.

We tried to improve the readability of the figure 1 by rearranging the images and removing the table.

  1. Higher magnification of ATO NPs should be provided in Fig. 1a. What is the scale bar of Fig. 1a and 1b?

The scale bar in figures 1A and 1B corresponds to 20 nm as indicated now in the images and in the caption.

  1. What is synthetic seawater? The components of artificial seawater should be introduced in the Materials and methods section.

Reply:  The synthetic seawater composition has been added to the materials and methods section.

  1. In Figure 3, what did time point represent in x axis? Regarding the data presented in Figure 3, it would be beneficial for readers to have a clearer understanding of the time points referred to on the x axis. It is important to provide clear labeling of x axis.

Reply:  Time points numbering were substituted by the real time of sampling for better clarity.

  1. In line 138, 146,270 and 272, the number should be written in subscript.

Reply:  We apologise for the typos. There was a change in format when transforming the manuscript to pdf. We will re-check it.

  1. For in vivo study of the acute toxicity, the ATO NPs seemed less lethal to the zebrafish adults or embryos. Besides, more experiments should be conducted to confirm that point, such as morphological analysis, genetic profiles as well as histological analysis.

Reply:  ATO NPs were not lethal for any of the zebrafish ages. We agree with the reviewer in the fact that more studies should be conducted to investigate the mechanism of sub-lethal toxicity on the embryos. Histopathological analysis and genomic, proteomic and transcriptomic analysis can be performed, now, with a more careful focus based on the endpoints that showed sensitivity to the presence of the NPs.

Comments on the Quality of English Language

The manuscript should be carefully checked and revised to avoid the expression, grammar and spelling errors.

Reply:  The manuscript has been carefully checked for expression, grammar and spelling errors.

Reviewer 4 Report

This is an interesting manuscript on an important matter.please see some comments and recommendations. when resubmitting please answer the comments in a .doc document that shows also where in the amended manuscript the changes are found

1) there is a need for english language corrections. Please read by a fluent English speaker and correct accordingly

2) in the introduction you should conclude with a paragraph clearly stating

a) what the environmental problem is b) what you plan to achieve with your research c) why this research is fit for an international audience

3) it has to be better justified how the two different domains of experiments that you performed link to each other-the bioaccumulation in mussels and the danio rerio effects. why these two models were chosen in particular? what questions they answer? this has to be clear in both abstract and discussion

4) in materials and methods for all chemicals and apparatuses used you have to give model if available, manufacturer, city and country of origin. this means for example all reagents used for the saltwater, all analytical apparatuses, oven, centrifuge, microscope, glass plates etc

5) some minor mistakes should be corrected such as Mytilus galloprovincialis and all latin names shoud be given in italics in both text and in references. Also some mistakes such as H2O etc should be subscripts. please correct throughout

6) It is not clear to me why you used ANOVA, ANCOVA and two-factor? ANOVA. what were the differences in each measured parameter that led to the differences in statistical analysis

7) I am not sure that the statistical differences eg in the form of different letters on the bars are shown in the graphs and figures

8) for so many results I believ that more references should be added in the discussion, especially on the importance of mussels as sentinel species. Please see and cite if possible Kasiotis et al 2015 Environmental Chemistry Letters Volume 13, Issue 4, Pages 395 - 411

9) There are mistakes in the reference list that have to be corrected. eg all latin names should be in italics , the year placement in ref no 49 is wrong, ref no 37 should be corrected, chemical formulas in titles should be corrected

minor english language correction needed

Author Response

This is an interesting manuscript on an important matter. Please see some comments and recommendations. When resubmitting please answer the comments in a .doc document that shows also where in the amended manuscript the changes are found

  • there is a need for english language corrections. Please read by a fluent English speaker and correct accordingly

Reply:  The manuscript has been carefully checked for expression, grammar and spelling errors.

2) in the introduction you should conclude with a paragraph clearly stating

  1. a) what the environmental problem is b) what you plan to achieve with your research c) why this research is fit for an international audience

ATO NPs have been used for different applications, in particular, incorporated in polymers to improve their optical properties and resistance to scratches. Their heavy metal content as well as their nanosize, makes them potentially hazardous materials for humans and the environment.  Despite being produced at significant levels and incorporated in commercial products, the ecotoxicity of those nanoparticles has been disregarded. Tin antimony oxide nanoparticles' ecotoxicity and bioaccumulation in aquatic organisms are described in this manuscript for the first time. This paragraph has been added at the end of the introduction section.

3) it has to be better justified how the two different domains of experiments that you performed link to each other-the bioaccumulation in mussels and the danio rerio effects. why these two models were chosen in particular? what questions they answer? this has to be clear in both abstract and discussion

Water resources are considered one of the main sinks for nanomaterials in the environment [31]. In this study, we wanted to make a preliminary assessment of the potential impact of ATO NPs when reaching water environment (both freshwaters and marine waters) using two highly sensitive organisms that can provide insights on the acute toxicity and bioaccumulation potential of NPs; zebrafish and marine mussels, respectively.

Zebrafish model is a demonstrated valuable tool for environmental health researchers evidenced by the steadily increasing number of studies that uses it (10.1016/bs.ctdb.2016.10.007).

The zebrafish embryo toxicity assay is a well-established and extensively used test in the environmental sciences, validated by the OECD, to assess acute aquatic toxicity by developmental defects resulting from exposure to environmental chemicals and (nano)particles. Additionally, those assays provide a simple, efficient and high-throughput toxicity testing in systems significantly more complex than cultured cells, and at the same time reducing the potential suffering of higher organisms, such as mice or rats.

On the other hand, due to their wide distribution and tendency for bioaccumulation, marine shellfish, especially mussels, have been suggested as suitable bioindicators for monitoring various toxic contaminants (metals and organic pollutants). Recent studies, such as the one by Sun et al. (https://doi.org/10.1016/j.jhazmat.2021.127383) found between 2.1 × 106 and 8.4 × 106 particles/mg of Ti-containing NPs in marine mussels, which contributed about half of the total Ti in various marine shellfish, suggesting that marine shellfish may be a significant sink for Ti-containing NPs.

Part of this text has been included in the introduction and discussion sections of the manuscript.

4) in materials and methods for all chemicals and apparatuses used you have to give model if available, manufacturer, city and country of origin. this means for example all reagents used for the saltwater, all analytical apparatuses, oven, centrifuge, microscope, glass plates etc

Model and manufacturer were included for instruments. City and country of origin were added when possible to retrieve. Reagents used for the artificial seawater were added in materials and methods section.

5) some minor mistakes should be corrected such as Mytilus galloprovincialis and all latin names should be given in italics in both text and in references. Also some mistakes such as H2O etc should be subscripts. please correct throughout

Reply:  We apologise for the typos. There was a change in format when transforming the manuscript to pdf. We will re-check it.

6) It is not clear to me why you used ANOVA, ANCOVA and two-factor? ANOVA. what were the differences in each measured parameter that led to the differences in statistical analysis

Factorial ANOVA is used to compare and contrast the means of two or more populations. ANCOVA is used to compare one variable in two or more populations while considering other variables.The primary difference among these statistical analysis is that ANCOVA is used when there are covariates (denoting the continuous independent variable), and ANOVA is appropriate when there are no covariates.

ANCOVA is a general linear model that combines ANOVA and regression analysis. ANCOVA evaluates whether the means of a dependent variable (e.g. yolk volume; pupil volume - respectively) are equal across levels of a categorical independent variable (treatment), while statistically controlling for the effects of other continuous variables that are not of primary interest, known as covariates (e.g. egg volume; eye volume - respectively) or nuisance variables. Mathematically, ANCOVA decomposes the variance in the dependent variable into variance explained by the covariates, variance explained by the categorical IV, and residual variance. In basic terms, the ANCOVA examines the influence of an independent variable on a dependent variable while removing the effect of the covariate factor.

7) I am not sure that the statistical differences eg in the form of different letters on the bars are shown in the graphs and figures

Statistical differences can be represented either by using different letters or different symbols; same letters or symbols indicate "no differences among treatments", while different letter or symbols corresponds to "significant statistical differences among treatments"; treatments without significant statistical differences do not have any letter or symbol.

8) for so many results I believe that more references should be added in the discussion, especially on the importance of mussels as sentinel species. Please see and cite if possible Kasiotis et al 2015 Environmental Chemistry Letters Volume 13, Issue 4, Pages 395 – 411

We have added some more references to the manuscript, including the one suggested, to reinforce the importance of mussels as sentinel species.

9) There are mistakes in the reference list that have to be corrected. eg all latin names should be in italics , the year placement in ref no 49 is wrong, ref no 37 should be corrected, chemical formulas in titles should be corrected

Reply:  We apologise for the typos. References were checked and corrected in the manuscript when needed.

Round 2

Reviewer 2 Report

The revised MS is much better than last version. But there are some aspects should be modified before acceptance.

1. The authors should pay attention to the use of abbreviations, e.g. line 28 sp-ICP-MS, please provide its full name. 

2. Line 49, ATO NPs here is redundant.

3.  Line 134 ® should be at the top right-hand corner.

4. Fig. 2 A should be re-drawn. It appears to be a series of garbled text.   The author should appropriately adjust the range of the Y-axis to ensure a proper layout.

5. The authors should re-review your MS, as there may be several formatting errors that have not been addressed yet.

English expression is fine.

Author Response

The revised MS is much better than last version. But there are some aspects should be modified before acceptance.

  1. The authors should pay attention to the use of abbreviations, e.g. line 28 sp-ICP-MS, please provide its full name.

Abbreviations’ use has been revised along the manuscript and modifications have been done where inconsistencies where found.

  1. Line 49, ATO NPs here is redundant.

Indeed there were redundancies in the sentences of the paragraph. We modified it accordingly to improve readability.

  1. Line 134 ® should be at the top right-hand corner.

This formatting error has been corrected.

  1. Fig. 2 A should be re-drawn. It appears to be a series of garbled text. The author should appropriately adjust the range of the Y-axis to ensure a proper layout.

We eliminated the text referring the statistical analysis as no significant differences were found for graph 2A. We have also adjusted the Y-axis range to the range of the data.

  1. The authors should re-review your MS, as there may be several formatting errors that have not been addressed yet.

We made a review of the full manuscript, and corrected the formatting errors found.

Reviewer 3 Report

According to the reviewers' comments, the manuscript has been carefully revised and greatly improved. Thus, I recommend the acceptance in the journal. 

Author Response

According to the reviewers' comments, the manuscript has been carefully revised and greatly improved. Thus, I recommend the acceptance in the journal.

We thank the reviewer for her/his positive comment.